# Epidemiological and clinical features of Panton-Valentine Leukocidin positive *Staphylococcus aureus* bacteremia: A case-control study

**Ming Da Qu**[1]*, **Humera Kausar**[1,2], **Stephen Smith**[2,3], **Peter G. Lazar**[4], **Aimee R. Kroll-Desrosiers**[4,5], **Carl Hollins III**[4], **Bruce A. Barton**[4], **Doyle V. Ward**[2,6], **Richard T. Ellison III**[1,6]

1 Department of Medicine, UMass Chan Medical School, Worcester, MA, United States of America, 2 Center for Microbiome Research, UMass Chan Medical School, Worcester, MA, United States of America, 3 Philips Healthcare North America, Andover, Massachusetts, United States of America, 4 Department of Population and Quantitative Health Sciences, UMass Chan Medical School, Worcester, MA, United States of America, 5 VA Central Western Massachusetts Healthcare System, Leeds, Massachusetts, United States of America, 6 Department of Microbiology and Physiological Systems, UMass Chan Medical School, Worcester, MA, United States of America

* mdq1990@gmail.com

**Data Availability Statement:** All files are available from the NCBI Bio Project database https://dataview.ncbi.nlm.nih.gov/object/PRJNA742741?reviewer=lstthqmepfaeag4fsc54554e1v.

## Abstract

### Background

Panton-Valentine Leukocidin (PVL) toxin in *Staphylococcus aureus* has been associated with both severe pneumonia and skin and soft tissue infections. However, there are only limited data on how this virulence factor may influence the clinical course or complications of bacteremic *S. aureus* infections.

### Methods

Between September 2016 and March 2018, *S. aureus* isolates from clinical cultures from hospitals in an academic medical center underwent comprehensive genomic sequencing. Four hundred sixty-nine (29%) of 1681 *S. aureus* sequenced isolates were identified as containing the genes that encode for PVL. Case patients with one or more positive blood cultures for PVL were randomly matched with control patients having positive blood cultures with *lukF/lukS-PV* negative (PVL strains from a retrospective chart review).

### Results

51 case and 56 control patients were analyzed. Case patients were more likely to have a history of injection drug use, while controls more likely to undergo hemodialysis. Isolates from 78.4% of case patients were methicillin resistant as compared to 28.6% from control patients. Case patients had a higher incidence of pneumonia and skin and soft tissue infection and longer duration of fever without differences in length of bacteremia. Clinical cure or expiration was comparable.

**Funding:** The authors received no specific funding for this work.

**Competing interests:** The authors have declared that no competing interests exist.

## Conclusions

These results are consistent with prior observations associating the PVL toxin with both community-acquired MRSA strains as well as severe staphylococcal pneumonia. The presence of the PVL toxin does not appear to otherwise influence the natural history of bacteremic *S. aureus* disease other than in prolonging the duration of fever.

## Introduction

*Staphylococcus aureus* bacteremia (SAB) is a common and serious infection with an incidence of 10 to 30 per 100,000 person-years in the industrialized world and a 10–50% case fatality rate among [1]. Both host and pathogen factors likely contribute to the development of SAB, and contributing host factors including age, male gender, ethnicity, injection drug use, hemodialysis, the presence of intravenous catheters, neutrophil dysfunction, iron overload, diabetes, cancer, corticosteroid therapy, and *S. aureus* colonization.[1–5]

Pathogen factors contributing to bacteremia have been less well defined, although both methicillin-resistance and the presence of the Staphylococcal *enterotoxin P* gene in methicillin-resistant *S. aureus* (MRSA) strains have recently been identified as risk factors for SAB. [2, 6] Of note, there has been a significant increase in the relative incidence of infections due to the community associated MRSA strain ST8-USA300 in North America [7] and Europe [1]. For instance, in Geneva, ST8-USA300 has been linked with transmission from the Americas [8]. An important virulence factor of this strain is the Panton-Valentine Leukocidin (PVL), a binary exotoxin, encoded by *lukF-PV* and *lukS-PV*, that forms pores in leukocyte membranes. [9–11] First identified in 1932, the toxin was initially associated with the development of abscesses, and more recently with severe, necrotizing pneumonia whether present in methicillin-susceptible or MRSA isolates [9, 12–16]

Given the association of PVL with these severe infections, it would seem possible that the presence of the toxin would impact the clinical manifestations of SAB. However, the limited available literature has not found significant differences in clinical outcomes (e.g., mortality) or severity (e.g., sepsis, septic shock) between PVL positive bloodstream isolates and PVL negative isolates, although some studies suggest an association between PVL and prolonged duration of bacteremia, and greater sites of metastatic infection.[2, 17–20]

The purpose of this study was to further assess the potential impact of PVL on the clinical characteristics of SAB comparing the clinical manifestations of bacteremic episodes due to *S. aureus* isolates that did or did not have the *lukF-PV* and *lukS-PV* genes encoding PVL.

## Methods

This study was approved by the University of Massachusetts Medical School institutional review board H00014429_2. Consent was not obtained as this is a retrospective study deemed minimal risk to subjects. Requirement for informed consent was waived by the UMass Chan Medical School IRB.

Between September 2016 and December 2019, a convenience sample of 1681 *S. aureus* isolates obtained from the UMass Memorial Medical Center clinical microbiology laboratory underwent comprehensive genomic sequencing as part of a proof-of-concept study on the utility of genomic sequencing in infection control surveillance [21].

To identify PVL-positive *S. aureus* isolates, WGS sequence data was first assembled (SPAdes [22]) and assembly metrics were applied to select isolates for inclusion (QUAST [23]; GC (%) between 30 and 33.5, contig N50 > 7500nt, # contigs (> = 1000 bp) < 500, and Total length (> = 1000 bp) between 2.5 and 3.0 Mbases. A BLAST database was created from all passing assemblies. A blastn [24] search against a data base created from all passing assemblies (parameters: -qcov_hsp_perc 80 -perc_identity 90) with a 1918nt query sequence containing the lukF-PV and lukS-PV genes (GenBank Accession CP039167; region 1513198..1515115) identified isolates as lukSF positive (PVL+).

For isolates included in the study, a core, single nucleotide variant (SNV) phylogeny was constructed. Variant calling was performed using Freebayes version v1.2.0-dirty [25] as deployed in the snippy pipeline, version 4.3.8 [26], using *Staphylococcus aureus subsp. aureus* NCTC 8325 (NC_007795.1) as reference. Custom scripts were used to determine core single nucleotide positions from the *snps.raw.vcf outputs and create a multi-fasta alignment. snp-sites [27] was then used to extract a core SNV alignment. FastTree2 (parameters: -gtr) version 2.1.10 [28] was used to generate a phylogenetic tree, which was then mid-point rooted.

Multi-locus sequence type assignments were determined from assemblies using mlst version 2.16.2 [29], making use of the PubMLST website (https://pubmlst.org/) developed by Keith Jolley [30] and sited at the University of Oxford. The development of that website was funded by the Wellcome Trust.

Case patients with one or more positive blood cultures for strains that were PVL+ were randomly matched with control patients having positive blood cultures for PVL-. A retrospective chart review was conducted via a review of the institution's electronic health records assessing for demographics (age, gender), risk factors for SAB, clinical symptoms, concomitant infections (criteria per S1 File), severity of illness, laboratory markers of severity, clinical cure, all-cause mortality, loss to follow-up, length of hospitalization, duration of fever, duration of bacteremia, duration of ICU stay, and maximum temperature. Patients required post-hospitalization follow-up to be considered a clinical cure. Furthermore, determination of outcome was made based on index hospitalization records and follow up notes related to the index hospitalization.

Categorical variables were compared with chi-square or Fisher's Exact tests. For continuous variables, Student's t-test with Satterthwaite adjustment, when appropriate, and Wilcoxon rank-sum tests for non-normal distributions were used. To further delineate the epidemiology of these isolates, they were classified according to methicillin resistance status as a subset of antimicrobial resistance profiles, as well as analyzed per multi-locus sequence type (MLST)

This study was approved by the University of Massachusetts Medical School institutional review board H00014429_2

## Results

From the pool of 1681 sequenced isolates, 56 patients with bacteremia due to PVL-negative *S. aureus* (controls) and 56 patients with bacteremia due to PVL-positive *S. aureus* (cases) had undergone analysis. Three of these case patients were removed due to multiple strains of *S. aureus* found in blood culture. Twelve control isolates and two case isolates that failed to meet assembly criteria were removed, and replaced randomly with twelve other control isolates, leading to a final analysis of 51 case patients matched with 56 randomly selected control patients. If a patient had multiple hospitalizations with bacteremia, the first encounter was chosen for analysis. The 107 case-control patients were comparable in age and gender, as well as most risk factors for bacteremia (Table 1). However, case patients were more likely to have a history of injection drug use, while controls more likely to undergo hemodialysis or have had

**Table 1. Patient demographics by PVL′ Case/Control status.**

| | Case N = 51 | Control N = 56 | P-value 1 | P-value 2** |
|---|---|---|---|---|
| Age at Collected Date (Mean ± SD, Range) | 50.1 ± 23.3 (2.6–96.0) | 55.3 ± 21.9 (0.1–100.1) | 0.23 | 0.18 |
| Methicillin Resistant | 40(78.4%) | 16(28.6%) | <0.001 | |
| Risk Factors (N, %) | | | | |
| Diabetes | 5 (9.8) | 15 (26.8) | 0.05* | |
| IVDU″ | 24 (47.1) | 6 (10.7) | <0.001* | |
| Immunosuppressed | 7 (13.7) | 5 (8.9) | 0.54* | |
| Immunosuppressive Medications | 5 (9.8) | 8 (14.3) | 0.28* | |
| Indwelling IV catheters | 5 (9.8) | 13 (23.2) | 0.07* | |
| Hemodialysis | 2 (3.9) | 11 (19.6) | 0.02* | |

′Panton-Valentine Leukocidin.

″Intravenous Drug Use.

*P-values in P-value 1 column for categorical variables from chi-squared tests, unless indicated with an asterisk (*) where Fisher's Exact Test was used; for continuous variables, p-values from Student's t-test with Satterthwaite adjustment when appropriate.

**P-values in P-value 2 column from Wilcoxon rank-sum tests for non-normal distributions.

indwelling IV catheters. Isolates from 78% of case patients were methicillin resistant as compared to 29% from control patients.

Case patients were more likely to have complained of chest pain and had more diagnoses of pneumonia and skin/soft tissue infection, with no differences seen in the incidence of endocarditis, osteomyelitis, or septic arthritis (Table 2).

Creatinine and alkaline phosphatase were higher among controls compared to cases (Table 3).

The percentage of patients who were clinically cured or expired was comparable. No differences in length of stay nor illness severity (by markers such as sepsis or septic shock) had been

**Table 2. Patient symptoms and infections by PVL′ Case/Control status.**

| Signs & Symptoms (N, %) | Case N = 51 | Control N = 56 | P-value 1 | P-value 2** |
|---|---|---|---|---|
| Fever | 35 (63.4) | 41 (73.2) | 0.31 | |
| Chills | 8 (15.4) | 10 (17.8) | 0.80 | |
| Abdominal Pain | 8 (15.4) | 3 (5.4) | 0.11 | |
| Chest Pain | 12 (23.1) | 1 (2.0) | <0.001 | |
| Localized Swelling | 3 (5.7) | 0 (0) | 0.11* | |
| Cough | 4 (7.7) | 5 (8.9) | 0.99* | |
| Infection Class (N, %) | | | | |
| Pneumonia | 19 (36.5) | 9 (16.1) | 0.02 | |
| Endocarditis | 12 (23.1) | 7 (12.5) | 0.21 | |
| Osteomyelitis | 10 (19.2) | 12 (21.4) | 0.82 | |
| Septic arthritis | 3 (5.8) | 3 (5.4) | 0.72* | |
| Skin/soft tissue infection | 22 (42.3) | 13 (23.2) | 0.04 | |
| Focal hepatic/splenic/renal abscess | 1 (1.9) | 0 (0.0) | 0.48* | |

′Panton-Valentine Leukocidin.

*P-values in P-value 1 column for categorical variables from chi-squared tests, unless indicated with an asterisk (*) where Fisher's Exact Test was used; for continuous variables, p-values from Student's t-test with Satterthwaite adjustment when appropriate.

**P-values in P-value 2 column from Wilcoxon rank-sum tests for non-normal distributions.

**Table 3. Laboratory markers by PVL Case/Control Status.**

| Characteristic | Case<br>N = 51 | Control<br>N = 56 | P-value 1[*] | P-value 2[**] |
|---|---|---|---|---|
| White blood cell count (10^3 cells/μL), on date of culture (Mean ± SD, Range) | 13.5 ± 5.3 (4.9,28.3) | 14.0 ± 7.3 (3.1–31.1) | 0.65 | 0.0.97 |
| White blood cell count (10^3 cells/μL), maximum (Mean ± SD, Range) | 17. ± 7.1 (6.0–37.5) | 17.3 ± 7.5 (4.3–38.4) | 0.92 | 0.98 |
| Creatinine (mg/dL) (Mean ± SD, Range) | 1.4 ± 1.1 (0.3–5.9) | 2.2 ± 2.8 (0.3–16.7) | 0.06 | 0.10 |
| Alanine transaminase (U/L)(Mean ± SD, Range) | 77.0± 253.9 (5.0–1626) | 35.9 ± 41.6 (5.0–216.0) | 0.31 | 0.56 |
| Alkaline phosphatase (U/L) (Mean ± SD, Range) | 85.6 ± 35.5 (23.0–190.0) | 137.1 ± 104.4 (34.0–608.0) | 0.005 | 0.02 |
| T. Bili (mg/dL) (Mean ± SD, Range) | 1.4 ± 2.2 (0.2–11.0) | 3.7 ± 9.9 (0.2–54.8) | 0.19 | 0.12 |

found. Duration of fever was longer among cases, however, despite no difference in duration of bacteremia; controls had more positive prior cultures (Table 4).

The differences in proportions between cases and controls were tested among patients receiving vancomycin, vancomycin in the first five days, cefazolin or nafcillin, and cefazolin or nafcillin in the first five days using a chi-square test across the four treatment groups and one control group (Table 5). The number of cases receiving cefazolin or nafcillin were significantly less than the number of controls, $p_{adj}$ 0.004, 95% CI [-0.53,-0.14], adjusting for multiple comparisons. The days of antibiotic therapy among cases and controls were tested as counts using a negative binomial regression model with treatment groups entered as a set of dummy (0,1) variables with the control group as the reference. No statistical difference in days of therapy was found (all p-values > 0.05).

**Table 4. Clinical outcomes by PVL Case/Control Status.**

| Characteristic | Case N = 51 | Control N = 56 | P-value 1[*] | P-value 2[**] |
|---|---|---|---|---|
| Sepsis | 5 (9.8) | 17 (30.4) | 16 | |
| Septic Shock | 10 (19.6) | 11 (19.6) | | |
| Previous positive culture (Mean ± SD, Range) | 0.3 ± 0.6 (0.0–3.0) | 1.5 ± 3.5 (0.0–14.0) | 0.02 | 0.77 |
| Duration of the bacteremia, days (Mean ± SD, Range) | 3.4 ± 3.5 (1.0–15.0) | 3.0 ± 2.8 (1.0–16.0) | 0.57 | 0.73 |
| Fever (deg;C), T MAX (Mean ± SD, Range) | 39.0 ± 0.7 (37.3–41.1) | 38.9 ± 0.5 (38.1–40.0) | 0.91 | 0.64 |
| Fever duration, days (Mean ± SD, Range) | 4.8 ± 6.5 (0.0–29.0) | 2.2 ± 1.9 (1.0–7.0) | 0.04 | 0.03 |
| Length of hospital stay, days (Mean ± SD, Range) | 15.4 ± 15.1 (0.0–89.0) | 11.4 ± 7.7 (0.0–50.0) | 0.74 | 0.10 |
| Length of ICU stay, days (Mean ± SD, Range) | 4.0 ± 5.8 (0.0–21.0) | 3.6 ± 5.5 (0.0–22.0) | 0.72 | 0.91 |
| Death Date Known (N, %) | 8 (15.7) | 10 (17.9) | 0.97 | |
| Number of positive blood cultures (Mean ± SD, Range) | 4.8 ± 4.5 (1.0–20.0) | 4.6 ± 3.4 (0.0–18.0) | 0.72 | 0.61 |
| Outcome of Infection (N, %) | | | 0.59 | |
| Cured | 25 (49.0 | 33 (58.9) | | |
| Expired | 11 (21.6) | 10 (17.9) | | |
| Unknown | 15 (29.4) | 13 (23.2) | | |

[′]Panton-Valentine Leukocidin.

[*]P-values in P-value 1 column for categorical variables from chi-squared tests, unless indicated with an asterisk ([*]) where Fisher's Exact Test was used; for continuous variables, p-values from Student's t-test with Satterthwaite adjustment when appropriate.

[**]P-values in P-value 2 column from Wilcoxon rank-sum tests for non-normal distributions

**Table 5. Antibiotics given by Case/Control status***.

| | Case | Control | P-value* ‡ |
|---|---|---|---|
| Average total days of antibiotic therapy | 32.3 | 23.6 | 0.08‡ |
| Average number of different antibiotics patients received | 2.9 | 2.9 | - |
| Number of patients receiving vancomycin | 46 | 52 | 0.73* |
| Number of patients receiving vancomycin in first 5 days | 41 | 39 | 0.29 |
| Average Total days of vancomycin therapy | 14.5 | 8.4 | 0.07 ‡ |
| Number of patients receiving cefazolin or nafcillin | 12 | 32 | 0.004 adjusted |
| Number of patients receiving cefazolin or nafcillin in first 5 days | 8(15.7%) | 13(23.2) | 0.46 |

* The differences in proportions between cases and controls were tested among patients receiving vancomycin, vancomycin in the first five days, cefazolin or nafcillin, and cefazolin or nafcillin in the first five days using a chi-square test across the four treatment groups and one control group. The number of cases receiving cefazolin or nafcillin were significantly less than the number of controls, padj 0.004, 95% CI [-0.53,-0.14], adjusting for multiple comparisons. The days of antibiotic therapy among cases and controls were tested as counts using a negative binomial regression model‡ with treatment groups entered as a set of dummy (0,1) variables with the control group as the reference. No statistical difference in days of therapy was found (all p-values > 0.05).

Phylogenetic analysis revealed that PVL-positivity was not randomly distributed amongst the patient cohort. *Staphylococcus aureus* ST8 strains were overwhelmingly represented among PVL+ isolates, whereas PVL- isolates had a distribution of sequence types, with the plurality being ST5 (Fig 1).

Subset analysis was performed to see if laboratory markers and clinical outcome differences (Table 6) were present between isolates from patients with a history of IVDU and those without. Though the primary outcome composite showed significant differences between IVDU and non-IVDU, the proportions of cured and expired illness, in light of the greater proportion of unknown outcomes among IVDU, were comparable.

Those with IVDU tended to be younger, with an equal number of men and women, and have far less prevalence of diabetes, indwelling IV catheters, hemodialysis, and immunosuppression (Table 7).

## Discussion

This retrospective, case-control study was conducted primarily to explore possible differences in clinical outcomes for SAB associated with the presence of PVL exotoxin. Secondarily, clinical and laboratory markers of severity, and epidemiologic data such as strains, comorbidities, and risk factors were compared.

Case patients were found to have longer duration of fever but there was no difference in duration of bacteremia, number of positive blood cultures, rates of cure, all-cause mortality, or rates of unknown outcomes between case and control isolates. Furthermore, there were no differences in duration of bacteremia or length of hospitalization. Deep-seated infections such as osteomyelitis, septic arthritis, and endocarditis were not found more often in the case patients, although there was approximately double the percentage of pneumonia and skin and soft tissue infections (SSTI) in case patients.

Differences in mortality (as part of the primary composite clinical outcome) were not seen in this case-control format study specifically evaluating patients with PVL-positive SAB against those with PVL-negative SAB. The data from other studies generally have not found a consistent difference in mortality, which is consistent with what has been usually observed in studies evaluating clinical outcomes and the presence of PVL.

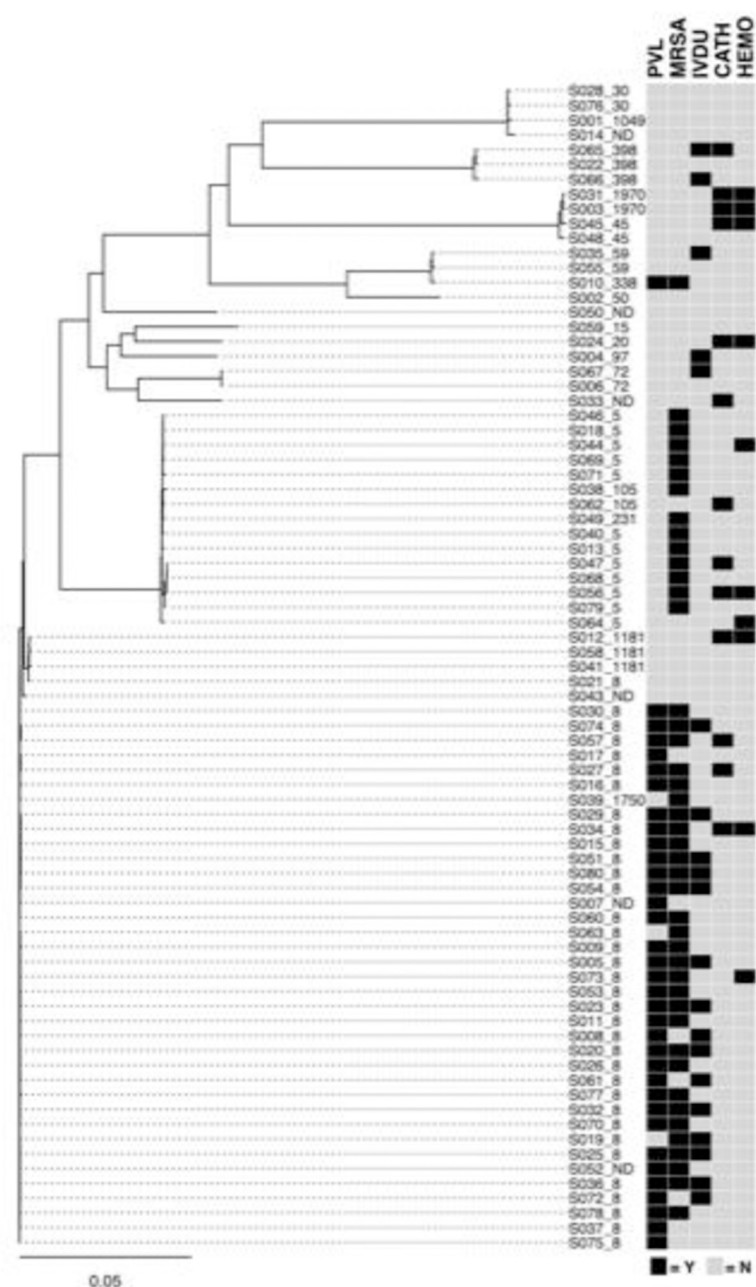

**Fig 1. *Phylogenetic analysis of* S. aureus *strains*.** Strains are identified by MicroSEQ ID_sequence type (ST). MRSA and PVL status along with selected clinical factors of interest are linked with strains. Legend: Strain/isolate numbers are listed as S###, followed by _ then sequence type (ST); PVL = Panton Valentine Leukocidin; IVDU = intravenous drug use; MRSA = methicillin resistant *Staphylococcus aureus;* CATH = presence of intravenous catheter; HEMO = on hemodialysis; Both PVL+ and PVL- strains included both methicillin-susceptible (MSSA) and methicillin-resistant (MRSA) strains with the PVL+ isolates predominately MRSA. A high percentage of patients infected with strains from the larger PVL-positive cluster had a history of intravenous drug use (IVDU).

One retrospective study in a Greek hospital [31] found no relationship between PVL and fatality in 32 isolates that spanned all types of invasive disease due to *S. aureus*. Another retrospective, 10-year study at a single center in the Gambia found that length of hospital stay and mortality were not associated with PVL status among 29 isolates from SAB [32]. Knudsen et al.

**Table 6. Patient outcomes by IVDU" History.**

| Characteristic | IVDU Yes N = 30 | IVDU No N = 77 | P-value 1 | P-value 2** |
|---|---|---|---|---|
| Sepsis | 2 (6.7) | 20 (26.0) | 0.41* | |
| Septic Shock | 4 (13.3) | 17 (22.1) | | |
| Previous positive culture (Mean ± SD, Range) | 0.8 ± 2.6 (0.0–14.0) | 0.9 ± 2.7 (0.0–14.0) | 0.30 | 0.84 |
| Number of positive blood cultures (Mean ± SD, Range) | 5.8 ± 4.9 1(0.0–20.0) | 4.2 ± 3.5 (1.0–18.0) | 0.14 | 0.21 |
| Fever (deg;C), T MAX (Mean ± SD, Range) | 39.1 ± 0.8 (37.9–41.1) | 38.5 ± 0.5 (37.3–40.1) | 0.29 | 0.39 |
| Fever duration, days (Mean ± SD, Range) | 4.1 ± 5.4 (0.0–25.0) | 3.1 ± 4.4 (0.0–29.0) | 0.48 | 0.26 |
| Length of ICU stay, days (Mean ± SD, Range) | 2.8 ± 5.4 (0.0–21.0) | 4.2 ± 5.7 (0.0–22.0) | 0.79 0.29 | 0.84 |
| Outcome of Infection | | | | |
| Cured | 11 (36.7) | 47 (61.0) | 0.002 | |
| Expired | 4 (13.3) | 17 (22.1) | | 0.003 |
| Unknown | 15 (50.0) | 13 (16.9) | | |
| Death Date Known (N, %) | 4 (3.7) | 14(13.1) | 0.75 | 0.77 |

"Intravenous Drug Use

*P-values in P-value 1 column for categorical variables from chi-squared tests, unless indicated with an asterisk (*) where Fisher's Exact Test was used; for continuous variables, p-values from Student's t-test with Satterthwaite adjustment when appropriate.

**P-values in P-value 2 column from Wilcoxon rank-sum tests for non-normal distributions.

found, among 129 PVL-positive SAB cases from a pool of 9490 total cases of SAB, no difference in all-cause 30-day mortality was seen in the unadjusted analysis [33]. Wehrhahn et al. found no difference in 30-day mortality when directly comparing PVL-positive with PVL-negative isolates from SAB [34].

On the contrary, Knudsen et al. in the same study found an association between higher 30-day mortality and PVL-positive status, after adjusting for illness severity and older age, and

**Table 7. Patient demographics by IVDU" history.**

| Characteristic | IVDU Yes N = 30 | IVDU No N = 77 | P-value 1 | P-value 2** |
|---|---|---|---|---|
| Age at Collected Date (Mean ± SD, Range) | 40.2 ± 13.8 (21.8–73.2) | 58.8 ± 23.7 (0.1–100.1) | < .0001 | < .0001 |
| Risk Factors (N, %) | | | | |
| Diabetes | 1 (3.3) | 19 (24.7) | 0.02 | |
| Immunosuppressive Medications | 0 (0.0) | 11 (14.3) | 0.02* | |
| Indwelling IV catheters | 2 (6.7) | 16 (20.8) | 0.14 | |
| Hemodialysis | 0 (0.0) | 13 (16.9) | 0.02* | |
| Genetic virulence factors (N, %) | | | | |
| PVL Positive (Case) | 24 (80.0) | 27 (35.1) | 0.00003* | |

"Intravenous Drug Use.

*P-values in P-value 1 column for categorical variables from chi-squared tests, unless indicated with an asterisk (*) where Fisher's Exact Test was used; for continuous variables, p-values from Student's t-test with Satterthwaite adjustment when appropriate.

**P-values in P-value 2 column from Wilcoxon rank-sum tests for non-normal distributions.

cited a 1.66 (1.16–2.38) adjusted hazard ratio [33]. Seybold et al. found a non-statistically significant lower crude in-hospital mortality in patients with SAB and PVL-positive status linked to USA300; mortality, however, was not seen to be lower in the univariate nor multivariate analyses. This study also did not find a difference in length of stay [35].

As there was a difference in the relative percentage of MRSA and MSSA strains between the cases and controls, a difference in the antibiotics used for SAB treatment would be anticipated and was found in our patient populations. As expected, case patients in our study had received more average days of vancomycin compared to controls, though more control patients had received vancomycin in the first five days. Despite the suggestion that vancomycin may be an inferior anti-staphylococcal antibiotic compared to semisynthetic penicillins or cephalosporins [36], case patients did not show worse outcomes compared to controls.

Case patients did tend to be younger and have IVDU compared to controls, though the subset analysis of IVDU did not suggest lower mortality associated with this risk factor. Many of these patients, compared to non-IVDU, had unknown follow up (not shown in data), rendering some degree of associated mortality and outcomes unknown.

The lack of difference in mortality may be in part attributable in part to bacterial genotypic/phenotypic factors. Hamilton et al. found varying levels of in vitro PVL toxin production from *S. aureus* isolates taken from patients with SSTI, pneumonia, and bacteremia; no correlation between disease severity and PVL production was found [37]. In a mouse model of *S. aureus* sepsis and skin abscess, infection with PVL genetic knockout strains compared to PVL+ strains did not produce differences in survival after bloodstream infection [38]. In fact, this study found that several PVL-negative strains were either more virulent or comparably lethal as genetically similar PVL positive strains, and that there was identical lysis of neutrophils in wild type and PVL knockout strains [38]. Badiou et al. found varying levels of PVL production *in vitro* according to sequence type, without a difference in production with regard to *mecA*-positivity. PVL+ isolates taken from patients with invasive infections spanning skin infections, pneumonia, and bone and joint infections were found to have 90% producing toxin concentrations deemed toxic to human leukocytes [39]. Other proteins related to virulence such as alpha-toxin, those regulated by the *agr* locus, the arginine catabolic mobile element (ACME) [1] have been implicated in virulence.

The predominance of MRSA compared to MSSA among the cases mirrors that found in a longitudinal, multi-year sample of about 1000 American isolates from multiple body sites, with PVL prevalence associated with MRSA being about 8 times that of MSSA, a trend seen across SSTI, bacteremia, lower respiratory tract infection, and other infections. [40] In that study, almost all isolates, regardless of methicillin resistance, were of the CC8 MLST, a combination identified as strain USA300. However, there were few bloodstream isolates. Case strains tended to be almost all ST8, compared to much greater heterogeneity of the control strains, with the plurality of the latter being ST5. This preponderance of ST8 among PVL-positive MRSA isolates reflects the successful dissemination of the USA300 MRSA strain in North America [1]; MSSA with the PVL locus have been characterized as highly diverse [41], though with genetic similarity to MRSA with PVL as well [41–43]. It has been theorized that PVL-positive CA-MRSA have arisen from PVL-positive MSSA strains [41, 44].

Strengths of this study include a case-control perspective focused on PVL exposure in SAB, the relatively high number of unique isolates, the WGS tracking of strain types. Some studies have explored PVL and clinical outcomes in the context of all invasive staphylococcal disease, with SAB as only a subset. Even among the studies that specifically evaluate PVL status and SAB, this study further evaluates clinically relevant parameters such as duration of fever, duration of bacteremia, and length of stay. Limitations include the retrospective study design, the relatively overall small sample size, the relatively high proportion of patients with unclear

outcome of infection, analysis of patients from only a single academic medical center, the use of multiple different antibiotic regimens for SAB treatment, and a relatively high number of patients with active IVDA which could impact patient follow up as well as outcomes including hospital LOS. treat.

This retrospective study examining differences in risk factors and clinically relevant outcomes (ie: sepsis, death, duration of bacteremia, length of stay, duration of fever) among patients with SAB who had infection due to *S. aureus* isolates with and without PVL found only a longer duration of fever without differences in length of stay, duration of bacteremia, nor degree of bacteremia.

## Supporting information

**S1 File. Supplementary material.**
(DOCX)

## Author Contributions

**Conceptualization:** Bruce A. Barton, Richard T. Ellison III.

**Data curation:** Humera Kausar, Stephen Smith, Peter G. Lazar, Doyle V. Ward.

**Formal analysis:** Ming Da Qu, Humera Kausar, Aimee R. Kroll-Desrosiers, Carl Hollins III, Bruce A. Barton, Doyle V. Ward, Richard T. Ellison III.

**Funding acquisition:** Doyle V. Ward, Richard T. Ellison III.

**Investigation:** Ming Da Qu, Humera Kausar, Richard T. Ellison III.

**Methodology:** Humera Kausar, Stephen Smith, Richard T. Ellison III.

**Project administration:** Humera Kausar, Richard T. Ellison III.

**Resources:** Humera Kausar, Doyle V. Ward, Richard T. Ellison III.

**Software:** Peter G. Lazar, Doyle V. Ward.

**Supervision:** Richard T. Ellison III.

**Validation:** Humera Kausar, Peter G. Lazar.

**Visualization:** Humera Kausar, Doyle V. Ward.

**Writing – original draft:** Ming Da Qu, Humera Kausar, Aimee R. Kroll-Desrosiers.

**Writing – review & editing:** Ming Da Qu, Humera Kausar, Aimee R. Kroll-Desrosiers, Bruce A. Barton, Doyle V. Ward, Richard T. Ellison III.

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
