## [Decision Letter · Decision Letter 0]

12 Oct 2021

PONE-D-21-28267Epidemiological and clinical features of Panton-Valentine Leukocidin positive Staphylococcus aureus bacteremia: a case-control studyPLOS ONE

Dear Dr. Qu,

Thank you for submitting your manuscript to PLOS ONE. After careful consideration, we feel that it has merit but does not fully meet PLOS ONE’s publication criteria as it currently stands. Therefore, we invite you to submit a revised version of the manuscript that addresses the points raised during the review process. In particular, reviewer 1 thinks more information is needed for "the selection of the case and control group," and tables need to be improved. Therefore, I recommend you revise your manuscript by addressing those review comments.

We look forward to receiving your revised manuscript.

Kind regards,

Taeok Bae

Academic Editor

PLOS ONE

Journal Requirements:

2. In your ethics statement in the manuscript and in the online submission form, please ensure that you have discussed whether all data/samples were fully anonymized before you accessed them and/or whether the IRB or ethics committee waived the requirement for informed consent. If patients provided informed written consent to have data/samples from their medical records used in research, please include this information.

Reviewers' comments:

Reviewer's Responses to Questions

**Comments to the Author**

1. Is the manuscript technically sound, and do the data support the conclusions?

Reviewer #1: No

Reviewer #2: Yes

2. Has the statistical analysis been performed appropriately and rigorously? 

Reviewer #1: No

Reviewer #2: I Don't Know

3. Have the authors made all data underlying the findings in their manuscript fully available?

Reviewer #1: No

Reviewer #2: Yes

4. Is the manuscript presented in an intelligible fashion and written in standard English?

Reviewer #1: Yes

Reviewer #2: Yes

5. Review Comments to the Author

Reviewer #1: The authors performed a retrospective case-control study to compare clinical characteristics and outcome of PVL+ and PVL- S. aureus bacteremia. The premise of the study is interesting, since PVL has been associated with SSTI and pneumonia but its relevance in S. aureus bacteremia (SAB) is still unclear. In total, 52 case and 56 control patients were included and analyzed. The authors concluded that the clinical cure was comparable and that the presence of PVL does not have any significant effect on the clinical course of SAB.

In order to assess the methodological quality of the paper, some missing information needs to be provided/clarified.

Major comments:

• In the selection of the case/control, was the antibiotic taken into account? The (optimal) treatment for MRSA SAB and MSSA SAB are not the same. MRSA=vancomycin and MSSA=beta-lactams/oxacillin. That being said, the proportion of MRSA/MSSA in both case and control groups were unclear. This information (and statistics) should be included in the tables. This is an essential information as vancomycin has been demonstrated to be inferior in terms of clinical cure (which is the outcome in the study) than oxacillin for treating MSSA SAB.

• In the US, most PVL+ strains are MRSA (ST8 USA300). Therefore, PVL- MRSA controls would have been the best choice of control group (or at least equal proportions of MRSA/MSSA). In the study 77% of the cases were MRSA and only 29% of the control was MRSA. Did all patients in both groups receive the same antibiotic treatment (vancomycin)?

• Tables are not intuitive. For example, table 6. How were the categories for vancomycin determined? Were these vancomycin plasma levels? If yes, units are missing and please state so. I assume that N=number of patients (e.g. for Vancomycin categories). It was unclear how the numbers in the column should add up (Total IVDU users N=30; but VAN <0.5 N=27; 1 N=38 and 2 N=5 giving a total of N=70)? Similar to outcome of infection (16+4+16 does not add up to N=30/column total). Please check your tables and numbers. Missing values and others should be included in the tables.

• These points should be clarified before other aspects of the study can be assessed.

Minor points

• Line 58: gene in italics

• Line 62: please cite the correct papers. Papers cited did not show the increase of ST8-USA300 in the community worldwide! (e.g. PMID: 30315958, 34160741, 26884428, 26464204, 33636329, etc..)

• Line 67: please cite the correct papers. Pneumonia PMID: 10524952, SSTI/Abscesses PMID: 25753191.

• Line 100: bacterial genus/species in italics

• Line 156: explain the coding of vancomycin

• Lines 211: length of stay, fever and other clinical parameters can only be assessed if information on the antibiotic therapy was provided (choice of antibiotic has an effect on the clinical course)

Reviewer #2: In the manuscript “Epidemiological and clinical features of Panton-Valentine Leukocidin positive Staphylococcus aureus bacteremia: a case-control study” authors describe the possible differences in clinical outcomes for SAB associated with the presence and absence of the PVL exotoxin. The manuscript presents interesting data, it is well structured and the writing quality is good.

Minor comments:

1. Line 39: correct percentage of MRSA in case patients

2. Please correct the order of tables (table #6 is after the table #3, and #4 is after #6)

3. Line 243: correct spelling of “however”

6. PLOS authors have the option to publish the peer review history of their article (what does this mean?). If published, this will include your full peer review and any attached files.

Reviewer #1: **Yes: **Dennis Nurjadi

Reviewer #2: No

---

## [Author Response · Author response to Decision Letter 0]

3 Feb 2022

Dear Dr. Bae,

My coauthors and I would like to thank the editors and reviewers for their thoughtful review of our manuscript entitled “Epidemiological and clinical features of Panton-Valentine Leukocidin positive Staphylococcus aureus bacteremia: a case-control study”( PONE-D-21-28267). We have now made a number of revisions to the manuscript to address the reviewer’s concerns and have attached both a clean version of the revision as well as a red-lined version showing all the revisions made to the original text. 

In addition, we would like to address each of the concerns raised by the reviewers.

Reviewer #1

Major comments:

• In the selection of the case/control, was the antibiotic taken into account? The (optimal) treatment for MRSA SAB and MSSA SAB are not the same. MRSA=vancomycin and MSSA=beta-lactams/oxacillin. That being said, the proportion of MRSA/MSSA in both case and control groups were unclear. This information (and statistics) should be included in the tables. This is an essential information as vancomycin has been demonstrated to be inferior in terms of clinical cure (which is the outcome in the study) than oxacillin for treating MSSA SAB.

In our selection of controls we did not include an adjustment for antibiotic therapy given that this was a retrospective real world review and the patient’s received multiple different antibiotic regimens initially and had multiple different subsequent modifications in antibiotic regimens. We appreciate the reviewer’s comment on the importance of providing this information, and we have now added this data to table 1and to a new table providing information on the antibiotic therapy received by the patients.

• In the US, most PVL+ strains are MRSA (ST8 USA300). Therefore, PVL- MRSA controls would have been the best choice of control group (or at least equal proportions of MRSA/MSSA). In the study 77% of the cases were MRSA and only 29% of the control was MRSA. Did all patients in both groups receive the same antibiotic treatment (vancomycin)?

We appreciate the reviewer’s suggestion about matching cases based on whether the case isolate was either MRSA or MSSA. However, our initial study plan was to solely assess the impact of PVL alone independent of the presence or absence of the mecA gene. In addition, the predominant use of vancomycin as opposed to a beta-lactam agent for MRSA infections would potentially have led to a worse clinical outcome for patients infected with a PVL+ isolate. This was not the finding of the study, and consequently we have chosen not to reselect a new control population for the study or to restrict our analysis to the PVL-/MRSA strains in our original control population. However, we have now added a description of this issue to the discussion section of our manuscript on lines 249 – 256. 

• Tables are not intuitive. For example, table 6. How were the categories for vancomycin determined? Were these vancomycin plasma levels? If yes, units are missing and please state so. I assume that N=number of patients (e.g. for Vancomycin categories). It was unclear how the numbers in the column should add up (Total IVDU users N=30; but VAN <0.5 N=27; 1 N=38 and 2 N=5 giving a total of N=70)? Similar to outcome of infection (16+4+16 does not add up to N=30/column total). Please check your tables and numbers. Missing values and others should be included in the tables.

In re-reviewing the data in the tables to make certain that the results have been accurately reported we found that one there was one individual included in who had 

simultaneous PVL+ and PVL- S. aureus bloodstream isolates; and this case has now been excluded from the study. This change has led to minor adjustments in results.

Data on follow up details were removed, as were vancomycin mic values; neither of these sets of data was felt to be relevant to this analysis. 

Minor points

• Line 58: gene in italics This has been corrected

• Line 62: please cite the correct papers. Papers cited did not show the increase of ST8-USA300 in the community worldwide! (e.g. PMID: 30315958, 34160741, 26884428, 26464204, 33636329, etc..) Reviewing literature and adjusting citations

• Line 67: please cite the correct papers. Pneumonia PMID: 10524952, SSTI/Abscesses PMID: 25753191. Reviewing literature and adjusting citations 

• Line 100: bacterial genus/species in italics This has been corrected

• Line 156: explain the coding of vancomycin Vancomycin mic no longer included in study. 

• Lines 211: length of stay, fever and other clinical parameters can only be assessed if information on the antibiotic therapy was provided (choice of antibiotic has an effect on the clinical course) We have incorporated a discussion of the impact of antibiotic therapy on these outcomes, as well as a table. 

Reviewer #2

In the manuscript “Epidemiological and clinical features of Panton-Valentine Leukocidin positive Staphylococcus aureus bacteremia: a case-control study” authors describe the possible differences in clinical outcomes for SAB associated with the presence and absence of the PVL exotoxin. The manuscript presents interesting data, it is well structured and the writing quality is good.

We thank the reviewer for his comments

Minor comments:

1. Line 39: correct percentage of MRSA in case patients this has been corrected

2. Please correct the order of tables (table #6 is after the table #3, and #4 is after #6) this has been corrected

3. Line 243: correct spelling of “however” this has been corrected

---

## [Decision Letter · Decision Letter 1]

3 Mar 2022

Epidemiological and clinical features of Panton-Valentine Leukocidin positive Staphylococcus aureus bacteremia: a case-control study

PONE-D-21-28267R1

Dear Dr. Qu,

We’re pleased to inform you that your manuscript has been judged scientifically suitable for publication and will be formally accepted for publication once it meets all outstanding technical requirements.

Kind regards,

Carla Pegoraro

Division Editor

PLOS ONE

Additional Editor Comments (optional):

Reviewers' comments:

Reviewer's Responses to Questions

**Comments to the Author**

1. If the authors have adequately addressed your comments raised in a previous round of review and you feel that this manuscript is now acceptable for publication, you may indicate that here to bypass the “Comments to the Author” section, enter your conflict of interest statement in the “Confidential to Editor” section, and submit your "Accept" recommendation.

Reviewer #1: All comments have been addressed

Reviewer #2: All comments have been addressed

2. Is the manuscript technically sound, and do the data support the conclusions?

Reviewer #1: Yes

Reviewer #2: Yes

3. Has the statistical analysis been performed appropriately and rigorously? 

Reviewer #1: Yes

Reviewer #2: N/A

4. Have the authors made all data underlying the findings in their manuscript fully available?

Reviewer #1: Yes

Reviewer #2: Yes

5. Is the manuscript presented in an intelligible fashion and written in standard English?

Reviewer #1: Yes

Reviewer #2: Yes

6. Review Comments to the Author

Reviewer #1: thank you for addressing the reviewers' comments, no further comments or concerns for the revised manuscript from my side.

Reviewer #2: (No Response)

7. PLOS authors have the option to publish the peer review history of their article (what does this mean?). If published, this will include your full peer review and any attached files.

Reviewer #1: No

Reviewer #2: No

---

## [Editor Report · Acceptance letter]

10 Mar 2022

PONE-D-21-28267R1 

Epidemiological and clinical features of Panton-Valentine Leukocidin positive *Staphylococcus aureus* bacteremia: a case-control study 

Dear Dr. Qu:

I'm pleased to inform you that your manuscript has been deemed suitable for publication in PLOS ONE. Congratulations! Your manuscript is now with our production department. 

Kind regards, 

on behalf of

Dr Carla Pegoraro 

Staff Editor

PLOS ONE